# Should We Be Trained to Train? Nursing Students’ and Newly Qualified Nurses’ Perception on Good Lecturers and Good Clinical Preceptors

**DOI:** 10.3390/ijerph16244885

**Published:** 2019-12-04

**Authors:** José Manuel Martínez-Linares, Celia Parra-Sáez, Carlos Tello-Liébana, Olga María López-Entrambasaguas

**Affiliations:** 1Department of Nursing, Universidad de Jaén, 23071 Jaén, Spain; omlopez@ujaen.es; 2Servicio Andaluz de Salud, 23007 Jaén, Spain; celiapasaez@gmail.com (C.P.-S.); carlostelloliebana@hotmail.com (C.T.-L.)

**Keywords:** educational nursing research, nursing students, preceptorship, nursing faculty practice, evaluation studies, qualitative research

## Abstract

*Background*: The reform of the Spanish higher education studies from the Bologna Declaration did not entail the necessary changes in the teaching methodologies used. The clinical preceptor emerged as the main guiding professional in the practical training of nursing students. The aim of this qualitative study was to understand fourth-year nursing students’ and newly qualified nurses’ (NQNs) perception on their lecturers’ and clinical preceptors’ effectiveness. *Methods*: Exploratory, descriptive qualitative study was carried out at a Spanish University. By convenience sampling and according to defined inclusion and exclusion criteria, twelve newly qualified nurses and twelve fourth-year students of the Degree in Nursing were included in order to contrast the results. A thematic analysis of data was carried out, to later be coded by two researchers. *Results*: Two main themes were identified: the good lecturer and the good clinical preceptor, with several subthemes in each. These included the characteristics that both should have, both in teaching, nursing and interpersonal-relation skills. *Conclusions*: The need of preceptorship training programs has been highlighted in our context. Educators all over the world should be properly qualified in order to train and educate competent nurses for the future.

## 1. Introduction

Universities are concerned with teaching effectiveness, and therefore, there is a need to address the quality of teaching and learning [1]. The classic teaching approach (teacher-centered) is changing worldwide into different new methods in order to increase students’ involvement. Problem-based learning, flipped learning, and game-based learning are some examples of new approaches based on the philosophy of student-centered learning [2].

The convergence of university teaching in the European Higher Education Area (EHEA) began upon the underwriting of the Bologna Declaration in 1999 [3]. Each Spanish University is empowered to design their own curricula. Since 2009–2010 academic year, Spanish universities have adopted a model of 90 ECTS (European Credit Transfer System) credits for clinical placements, divided into six ECTS credits for the end-of-degree project and 84 credits for supervised rotating placements, practical lectures in small groups, and autonomous work [4].

These changes in university teaching also led to changes in the way of teaching. For Zabalegui and Cabrera [5], the changes that nursing education programs have undergone include the shift from lecturer-centered to student-centered programs, as well as the shift from lectures as the main teaching methodology to a problem-based learning where students have to cooperate.

Arrogante [6] also argues that the modification of the nursing curricula to adapt to the EHEA is based on a set of competences that students must acquire throughout their training, along with some changes in the way lecturers teach. However, lectures are still overused without fostering critical thinking and autonomous work among students.

Practical training has been of great importance to professional nursing since nursing became part of university education in 1977. According to the European Directive on the recognition of professional qualifications [7], practical training will be carried out in hospitals and other health centers under the supervision of nursing teaching staff with the cooperation and assistance of other qualified nurses. In Spain, Order CIN/2134/2008 of July 3 [4], which establishes the requirements for the verification of official university degrees that qualify for nursing work, determines that this supervised practical training will be carried out through rotating internships with a final evaluation of competences in health centers, hospitals, and other healthcare centers, integrating the knowledge, skills, and attitudes of nursing into professional practice. To this end, Spanish universities reach agreements with public administrations and/or private health centers to create positive learning environments. The preceptor as a standard professional for students did not exist before, appearing from this new regulation. From that moment on, there was a person responsible for each student’s practical training. There is no international definition about what is a good or a bad lecturer and clinical preceptor. However, some authors have identified five categories of important qualities [8]: teaching abilities, nursing competence, evaluation, interpersonal relations, and personality. These are the categories included in the Nursing Clinical Teacher Effectiveness Inventory, which is the most used questionnaire regarding this topic. On the other hand, according to Kornell and Hausman [9], the better teachers were found to be the ones who contributed the most to learning in subsequent courses. “Deep learning” is the concept they use to refer to this kind of knowledge.

The nursing clinical preceptor has a fundamental role [10]. Billay and Yonge [11] defined clinical preceptors as experienced professionals who teach, supervise, and act as role models for students in degrees involving clinical placements. They must therefore have qualities such as work experience, leadership, collaboration, and consulting skills [12]. Other authors, however, address the need of prior training in teaching to perform such a function [13,14]. 

Several studies highlighted the benefits of preceptorship [15,16]. For Christiansen and Bell [15], preceptorship may improve students’ learning experience and their preparation as a future professional. For Moore and Cagle [16], support and socialization are necessary to work as nurse in a complex and multifaceted healthcare environment. However, what is “a good lecturer” and “a good clinical preceptor” to newly qualified nurses (NQNs) and nursing students? What qualities and characteristics should they have?

This paper is part of a larger project on the acquisition of nursing skills and the improvement of the academic curricula. One of the specific aims of this project is included here: to understand fourth-year nursing students’ and NQNs’ perception of their lecturers’ and clinical preceptors’ effectiveness.

## 2. Materials and Methods 

The methodology is common to all papers, and the breadth of the results is such that it cannot be included in a single paper. Part of them have been recently published [17]. However, this study on the characteristics that “a good teacher” and “a good tutor” should have led to plenty of sufficiently important results to generate an article by itself. The descriptive qualitative method used is presented following the criteria included in the consolidated criteria for reporting qualitative research format (COREQ) [18].

### 2.1. Research Team and Reflexivity

The research team consisted of two nursing educators and two research associates trained in qualitative research methodology, specifically focus groups. Interviews and focus groups were conducted by the four authors. The focus groups were organized by the associate researchers, after receiving instruction on the design and execution of this research technique in order to avoid any biased information that could be generated by the teaching staff when verifying the information provided about the training received.

None of the researchers knew or had any previous relationship with participants, who were previously informed about the project, the institution behind it, its objectives, form of participation, and the interviewers.

### 2.2. Study Design, Participant Selection, and Data Collection

A qualitative, exploratory, and descriptive research study was carried out. This was the most appropriate methodology due to the complexity of the subject and the difficulty in measuring the concepts [19]; moreover, that allowed participants’ subjective perception to be revealed [20]. Graduates of a Spanish university who wished to participate in the project were contacted by email in 2015, 2016, and 2017 for collaboration. The inclusion criterion was to have between 3 and 12 months of work experience as a healthcare professional in Spain. Participants were selected by convenience sampling. During the first phase of the study, 12 semi-structured personal interviews with NQNs were conducted. Nobody resigned from participating. With this number of interviews, data saturation was achieved [21]. 

During the second phase of the study, two focus groups were made up with 12 fourth-year students of the Degree in Nursing of this University to contrast data obtained in personal interviews with NQNs. The same approximation method was used and the people who were willing to participate were selected. None of them refused to do so during the later process. Table 1 shows their sociodemographic data.

In both phases, the data were collected in a room of the faculty, or else via Skype (Microsoft^©^) version 8 (Jaén, Spain) in case the interviewee was located far from the venue. All participants in the focus groups were present in the room. No one else was present.

Interview questions were drafted ad hoc after a previous review of the literature and had three thematic sections: theoretical training, clinical training, and working situation. Only questions about theoretical and clinical training were asked to answer the aim of this paper (Table 2). The interview and focus group were trialed with three nurses and former students of this university. Questions were modified according to the interviewees’ remarks about clarity, relevance, and importance. No interviews or focus groups had to be repeated, lasting 45–90 min and 60–90 min, respectively.

### 2.3. Data Analysis

Personal interviews and focus groups were recorded in audio for later transcription. Field notes were taken by the research team and incorporated as data for analysis.

The data analysis was carried out by two researchers, who jointly presented the important themes for analysis [22]. For the analysis of the transcripts, the six-phase method of thematic analysis with scientific precision by Braun and Clarke [23] and the process to ensure the reliability of the results were followed [24]. 

The coding was carried out independently by two researchers, generating a description of their respective coding trees and the relationship between the identified themes and their respective subthemes. An agreement was reached on the codes finally used, as well as on the themes and subthemes generated in the analysis (Table 3). The software Atlas.ti version 7 (Jaén, Spain) for Windows (Microsoft^©^) was used.

For the analysis of the focus group transcriptions, the contents were submitted to direct analysis. Previously established codes were used and applied to the new data obtained [25].

### 2.4. Ethical Considerations

The study was carried out by following the ethical principles of the Declaration of Helsinki. Personal data were processed in accordance with Regulation (EU) 2016/679 of the European Parliament and of the Council of April 27, 2016 on the protection of natural persons with regard to the processing of personal data and on the free movement of such data. 

Each participant in face-to-face interviews or focus groups was asked to complete the corresponding informed consent. The study was carried out after obtaining the approval of the ethics committee of this University.

## 3. Results

Two themes emerged: “the good lecturer” and “the good clinical preceptor”, which included several subthemes (Figure 1). Their perceptions can be grouped as follows:

### 3.1. Qualities Related to Teaching Skills

The “good lecturer” has to know how to motivate students to go in depth in the contents of courses and has to teach these contents in a dynamic way. In order to do this, lecturers should explore different teaching aids, far from merely and solely using slides.
“For me, a good lecturer is the one who encourages the group to learn, and they don’t just read slides. When they do that, you totally switch off.”(E2)
“He/She has to be able to motivate students and use new teaching methodologies.”(GF1-P2, GF1-P6)

### 3.2. Qualities Related to Nursing Competences

They must have previous healthcare work experience and must continuously update their training and teaching resources. It is perceived that previous healthcare experience makes it possible to reduce the theory–practice gap, to have a greater and better knowledge of the contents taught, and to use this experience didactically in a such a way that students feel that what they are learning is real.
“The way to catch attention by talking about situations that happened in your professional life and link them to the theory [...]. And you take it as a real-life situation to which you can relate the theory you’re going to learn.”(E4)
“He/she must have an academic background in Nursing, as well as a concern for continuing education and knowledge updating.”(GF1-P2, GF2-P7, GF2-P11)

### 3.3. Qualities Related to Interpersonal Relations and Personality

The undergraduate students highlighted qualities related to interpersonal relationships and the personality of the “good lecturer”, specially closeness and accessibility toward students, as well as having a good sense of humor and friendliness. This leads to better lecturer–student relations which enable to deal with other students’ concerns, and not just those related to academic issues.
“Closeness of lecturers towards us is also good, which is not just a lecturer-student relationship, but we can also talk about other topics that concern us or interest us.”(GF2-P8, GF2-P10)

### 3.4. Characteristics of a Clinical Preceptor

NQNs interviewed defined “the good preceptor” based on the strengths and weaknesses they noticed in their own clinical preceptors. The strengths were having a broad career, having motivation and involvement to carry out the preceptorship, and being approachable professionals who know how to organize themselves at work. All this generates security and confidence in students’ learning experience.
“Experience, more than anything else, the experience and security provided by the experience of years of work. And I would also say the organization... To know how to organize the daily work.”(E11, E12)
“As strength, the involvement and motivation to teach, to be with the students and to be approachable, because that shows they like their work.”(E5)

The weaknesses highlighted were the perception of boredom all around the preceptorships and the experience-based practice rather than the evidence-based practice along with reluctance to changes. All of this is detrimental to students’ learning in their rotating internships by generating discomfort and insecurity, as well as enhancing the perception of the theory–practice gap.
“There are a lot of preceptors who are fed up; they don’t want more students and don’t teach with enthusiasm or they just don’t explain things to you.”(E7)
“Professionals who have been working for many years and for whom the scientific evidence, the search for bibliography and so on..., yes, they appreciate all that, but little… though… little do they put it into practice. And they don’t intend to change either.”(E10)
For undergraduate students, “good preceptors” “do not feel obliged to supervise” (GF1-P5), “want to supervise and show devotion to teaching” (GF2-P8, GF2-P11), “involve themselves in supervision” (GF1-P6, GF2-P10), and “are nice people.”(GF2-P7)

### 3.5. The Importance of Socialization

Students and NQNs expressed that an introduction to the unit, staff, location of the material, organization, operation, type of patient being treated, and the main techniques carried out by clinical preceptors at the beginning of the student’s internship in a new hospital or primary care unit, makes students have the perception of belonging to a work team and their achievement is greater.
“[...] that, in the first day of a rotating internship, a person in charge, before starting working, shows you the medication room, the supply room, how the rooms are organized, staff, the way they work, the way they divide the work...” [...]. When it happened to me, I felt less lost, more integrated and more eager to learn and do things.”(GF1-P2)

## 4. Discussion

The characteristics and qualities of the “good lecturer” referred by the NQNs and the students of this study are similar to the features recorded in the teaching-ability subscale of the Nursing Clinical Teacher Effectiveness Inventory [8]. With this measuring tool, they highlighted the characteristics corresponding to the subscale of nursing competence: being a good role model, enjoying nursing, demonstrating clinical skills and experience, or being well-prepared to teach [26]. One study, in reply to the latter [27], highlighted the characteristics included in the subscale of interpersonal relationships, while the study conducted by Soriano and Aquino [28] obtained higher scores in the characteristics of the personality subscale: being an organized, self-confident, dynamic, and self-critical person. 

As for the characteristics of the “good clinical preceptor”, this study differs methodologically from previous ones [27,29], but there are similarities, even with those in which other methodologies were used. Sweet and Broadbent [30], despite achieving 9.5% of the sample participation, highlighted the following characteristics: availability and willingness to help, approachability and disposition, and provision of feedback to students on the activities carried out. The analysis of an open-ended question showed that the qualities of a “good preceptor” were motivation for tutoring, good work organization skills, dedication and involvement in tutoring, and kindness.

Approachability was also highlighted by Jansson and Ene [31] as a positive attitude because of the security transmitted to placement students. This, together with interpersonal skills, are the two characteristics that a preceptor must have to be effective, according to the conclusions highlighted in a recent integrative review [32]. According to the study by Niederriter et al. [33], both accessibility and availability of the clinical preceptor are important characteristics that enhance the learning experience, as they contribute to the establishment of a relationship of trust and a feeling of comfort. Previous healthcare experience was also highlighted, as it builds confidence among students; and good work organization skills, as clinical preceptors should be proven good work organizers to later try to boost such a behavior among students. 

The preceptorship programs have demonstrated their usefulness in overcoming the theory–practice gap thanks to the preceptor–student links established [34,35] in order to improve nursing teaching and practice quality [36]. Its results are translated into positive clinical experiences of the students that lead to better learning outcomes during their clinical placements [37]. Thus, training in up-to-date educational methodologies and continuous assessment [13] is necessary, which will also improve competence acquisition and student confidence [38]. However, neither Spanish universities nor Spanish public health systems have carried out relevant preceptorship training programs which could have been subjected to assessment. The only requirement to be a preceptor is to wish to be one without remuneration. This lack of training was highlighted by Hickey [39], in whose study 74% of the preceptors had not participated in a preceptorship training program.

The importance of socialization at the beginning of their clinical placements was also highlighted as an improvement factor in the learning process. Happel [40] included the role of the preceptor in the definition of preceptorship as a strategy to maximize the benefits of practical nursing training. Being the person in whom both NQN and students place their trust to socialize while performing their roles [39] has also been suggested as a responsibility inherent in preceptors [41], or as a dependency generated in the student toward the preceptor to achieve such socialization by immersing themselves in an environment which appears to be strange initially [42].

This study shows consistency with other studies [33] in which nursing students perceive that an effective preceptor has to be a role model who helps to achieve socialization in the health field as a learning environment.

Authors should discuss the results and how they can be interpreted in perspective of previous studies and of the working hypotheses. The findings and their implications should be discussed in the broadest context possible. Future research directions may also be highlighted.

### Implication for Practice

Spanish universities and public health systems need preceptorship training programs, which do not currently exist, with assessable objectives for both students of the Nursing Degree and clinical preceptors to be able to determine the benefits of such programs in preceptors’ function throughout their daily activity and in the effectiveness and quality of the students’ practical training.

## 5. Conclusions

The “good lecturer” has to know how to motivate students for learning, to master the contents of the courses, to know how to use innovative teaching tools that help to motivate students, to have previous healthcare experience, to update their training, and to be close and friendly.

The “good clinical preceptor” should have a broad career. They should also be motivated and involved in preceptorship and know how to organize their work. Tiredness about the idea of the preceptorship and reluctance to abandon experience-based practice would undermine students’ learning effectiveness. Besides, students perceived socialization at the beginning of placements as a good practice among preceptors.

The qualification of educators is a timely topic that should be taking into account internationally to provide best way to prepare the next generations of nurses.

## Figures and Tables

**Figure 1 ijerph-16-04885-f001:**
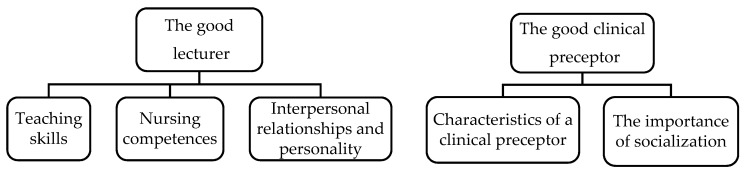
Themes and subthemes. Source: prepared by the authors.

**Table 1 ijerph-16-04885-t001:** Sociodemographic data of interviewees and participants of focus groups.

Personal Interviews					Focus Groups		
Interview Code	Age	Sex	Work Experience in Months	Graduation Year	Focus Group Code	Age	Sex
E1	26	Woman	3	2017	GF1-P1	40	Woman
E2	23	Woman	4	2016	GF1-P2	21	Man
E3	25	Woman	7.5	2016	GF1-P3	21	Man
E4	23	Woman	3	2016	GF1-P4	23	Woman
E5	23	Man	3	2016	GF1-P5	22	Woman
E6	25	Woman	3	2015	GF1-P6	24	Woman
E7	23	Woman	9	2016	GF2-P7	21	Man
E8	28	Woman	6	2017	GF2-P8	23	Man
E9	25	Woman	6	2016	GF2-P9	51	Man
E10	28	Man	9	2017	GF2-P10	21	Man
E11	25	Man	12	2015	GF2-11	21	Woman
E12	51	Man	12	2015	GF2-P12	22	Man

Source: prepared by the author.

**Table 2 ijerph-16-04885-t002:** Main questions of the interview guide.

Pre-Established Categories	Main Questions
Clinical training	How was your relationship with your clinical preceptors? Do you think your preceptors were adequately trained to train? Think about your favorite preceptor and tell me the reasons why she/he was your favorite. Think about the preceptor you most dislike and tell me why.
Theoretical/Academic training	What is your opinion about the teaching methods in class? Do you think your lecturers were qualified to teach theory? How was your relationship with your lectures?

Source: prepared by the author.

**Table 3 ijerph-16-04885-t003:** Themes, subthemes, and codes.

Themes	Subthemes	Codes
The good lecturer	What is a good lecturer?	Good lecturer, satisfaction with lecturers, well-prepared lecturer, lecturer’s experience, lecturers who become stagnated
	Improve teaching planning	Satisfaction with lecturers, satisfaction with preceptor(s), aspects of improvement in teaching
The good clinical preceptor	What is a good clinical preceptor?	Positive aspects of the preceptor, weaknesses of the preceptor(s), strengths of the preceptor(s), good preceptor, preceptors: appreciated qualities
	A wake-up call to clinical preceptors	Improving preceptors, well-prepared preceptors, negative aspects of the preceptor, preceptors: lack of motivation, wish to tutor

Source: prepared by the author.

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
