# Peer review of "Should We Be Trained to Train? Nursing Students’ and Newly Qualified Nurses’ Perception on Good Lecturers and Good Clinical Preceptors"

_ijerph, 2019, doi:10.3390/ijerph16244885_

Round 1

Reviewer 1 Report

Thank you for the opportunity to review this manuscript: “Should we be trained to train? A qualitative approach to good lecturers and good clinical preceptors.”

Originality/Novelty:  The research question (lines 15 & 71): The aim is to understand the newly-qualified nurses and 4th-year Nursing students about what a “good lecturer” and “a good clinical preceptor” is and what their characteristics and skills are. Needs revision. The title of the manuscript and the purpose of the research are disconnected. The title does not indicate that the qualitative approach is to seek the perceptions of nursing students and recent graduates as to what makes a good lecturer and a good preceptor. At first read, my expectation was that I would be reading research conducted with educators, not students. I suggest this be made clear from the beginning to better serve the reader. A “catchy” title is not always in the manuscript’s best interest.

The wording of the research question is also unclear, with poor sentence structure which may stem from an English as a second language issue. I believe the authors are trying to say in the abstract and in the body of the text: The aim of this qualitative study was to understand the perceptions of 4th year nursing students and recently graduated nurses of the characteristics of an effective lecturer and an effective clinical preceptor.     

Significance: The research question holds high significance to an international audience. Many researchers are currently looking at the best way to prepare the next generation of nurses and looking back to the question of preparation for the educators. Therefore, I see this as a timely topic, however some revisions are needed to improve the clarity and international appeal of this work. 

Quality of Presentation: The manuscript focuses on the changes in Europe and a specific Bologna Declaration that will not be familiar to many (or most) outside of this region. My suggestion is to begin the article in a more global tone, describing the global issue of best practice in teaching nursing students and then focus to the specific example of the Spanish experience as an example of attempts at change. Line 48 the use of the word “abused” is confusing to the reader. This should be expounded on and I would recommend finding a synonym for this word. Are the faculty abusing the lecturing process or are they reverting to a familiar mode of teaching that is within their comfort level? Could the use of lecture classes be a cost issue for nursing programs? The case for this could be made in the USA nursing programs with cost issues in higher education. Large lectures are less expensive for the universities. 

The paragraphs starting at Line 50 through Lind 70 were confusing to me. Is the training of nursing students in a hospital setting a new concept in Spain? I am not familiar with this as an educator in the US, this has been the standard of education for decades. The way this is written it sounds like this is a new undertaking with the EHEA that was introduced earlier. Or is the change that nurses now require university level degrees. This is something very different and not clear to the reader who is unfamiliar with the system and thus the setting for this research.   

Line 80 – 83 – The abbreviations are unknown to me. Common abbreviations such as RN and PhD are known but not RM, The use of the abbreviations of the author names is confusion. I would suggest describing the team more broadly such as: The research team consisted two nursing educators and two research associates trained in qualitative research methodology, specifically focus groups.

Line 91 – confusing sentence, perhaps this is an English language issue but need a rewrite for clarity. Suggest dividing to two sentences.

Line 93 – I would recommend removing the specific name of the university. It is not relevant and may create bias. A sentence clarifying that interviews were conducted with the newly graduated nurses and focus groups were conducted with the nursing students would be helpful to the reader.

Table 1: clear and easy to understand.  

Line 109 – Is this sentence saying that the interview and focus group questions were trialed with 3 nurses and former students? This was unclear to the reader.

Line 141 is redundant. Remove from above or here but does not need to be stated twice.

Figure 1 – (line 143) I was confused by the statement that two themes emerged identified as good lecturer and good clinical preceptor. Did these terms come out of the thematic analysis or were they part of the research aim to begin with? The aims should precede the research therefore the statement that these themes emerged from the research makes the aim statement suspect of post hoc manipulation by the research. Perhaps this is just a wording choice issue but needs addressed.  Additionally, it does not make sense that the emerging theme for a good clinical preceptor by the question “What is a good clinical preceptor?” please revise. 

Line 175 – What is NQN?

Line 208 – this is the first mention of the Nursing clinical teacher effectiveness inventory. Did you survey the sample or do interviews/focus groups?  Is this a quantitative study or mixed methods?   The first paragraph of the discussion section does not make sense to the reader.

The second paragraph of the discussion section does not seem to go with the study but speaks about the education requirements. If this is significant background information, I suggest moving it to the introduction. It loses the reader focus in the discussion. 

Line 271 – states the generally negative experiences with clinical preceptorships… This is the first time that the idea of negative experiences is introduced. The examples given from the data are not particularly negative and the tone of the manuscript does not identify a particularly negative experience. This needs to be revisited.

The conclusion needs to have a more global significance to the research. As it is written the research is only applicable to the Spanish universities and yet all nursing programs face many of these same challenges seeking qualified educators.

Author Response

Cover letter – Major revision - Reviewer 1

Thank you very much for the suggested modifications for the paper Should we be trained to train? A qualitative approach to good lecturers and good clinical preceptors. Having them already done, all the authors believe that the article has improved.

Every modification done is described below. In the paper you can find them in red colour:

Originality/Novelty:  The research question (lines 15 & 71): The aim is to understand the newly-qualified nurses and 4th-year Nursing students about what a “good lecturer” and “a good clinical preceptor” is and what their characteristics and skills are. Needs revision. The title of the manuscript and the purpose of the research are disconnected. The title does not indicate that the qualitative approach is to seek the perceptions of nursing students and recent graduates as to what makes a good lecturer and a good preceptor. At first read, my expectation was that I would be reading research conducted with educators, not students. I suggest this be made clear from the beginning to better serve the reader. A “catchy” title is not always in the manuscript’s best interest.

We have modified the title according to your recommendation, to make clearer the content of the paper.

The wording of the research question is also unclear, with poor sentence structure which may stem from an English as a second language issue. I believe the authors are trying to say in the abstract and in the body of the text: The aim of this qualitative study was to understand the perceptions of 4th year nursing students and recently graduated nurses of the characteristics of an effective lecturer and an effective clinical preceptor.

Thank you very much for this suggestion. We have modified the aim as suggested in the body of the text (lines 82-83) and in the abstract (lines 16-17). We think that now is clearer. We have maintained the term newly-qualified nurses because it is a term widely used in the literature.

Significance: The research question holds high significance to an international audience. Many researchers are currently looking at the best way to prepare the next generation of nurses and looking back to the question of preparation for the educators. Therefore, I see this as a timely topic, however some revisions are needed to improve the clarity and international appeal of this work. 

Thank you very much for your intention to provide an international appeal of this work. We have added a sentence in Conclusions section about it (lines 278-279). Also, modifications carried out in Introduction section try to provide an international appeal of this work (lines 31-35).

Quality of Presentation: The manuscript focuses on the changes in Europe and a specific Bologna Declaration that will not be familiar to many (or most) outside of this region. My suggestion is to begin the article in a more global tone, describing the global issue of best practice in teaching nursing students and then focus to the specific example of the Spanish experience as an example of attempts at change.

We have added a first paragraph in Introduction section in a more global tone, describing the best and newest practices in teaching nursing students (lines 31-35). Besides we have removed some content of the Introdcution section and the corresponding reference.

Line 48 the use of the word “abused” is confusing to the reader. This should be expounded on and I would recommend finding a synonym for this word. Are the faculty abusing the lecturing process or are they reverting to a familiar mode of teaching that is within their comfort level? Could the use of lecture classes be a cost issue for nursing programs? The case for this could be made in the USA nursing programs with cost issues in higher education. Large lectures are less expensive for the universities.

As you suggest, in line 48 we have replaced the term “abused” with the term “overused”.

The paragraphs starting at Line 50 through Lind 70 were confusing to me. Is the training of nursing students in a hospital setting a new concept in Spain? I am not familiar with this as an educator in the US, this has been the standard of education for decades. The way this is written it sounds like this is a new undertaking with the EHEA that was introduced earlier. Or is the change that nurses now require university level degrees. This is something very different and not clear to the reader who is unfamiliar with the system and thus the setting for this research. 

We have added two sentences at the beginning and end of the paragraph (lines 50-51 and 60-62). We think is much clearer that practical training and preceptor´s role changed due to the new European Directive.

Line 80 – 83 – The abbreviations are unknown to me. Common abbreviations such as RN and PhD are known but not RM, The use of the abbreviations of the author names is confusion. I would suggest describing the team more broadly such as: The research team consisted two nursing educators and two research associates trained in qualitative research methodology, specifically focus groups.

Nice. We have changed as suggested. Previously, we indicated names and degrees as COREQ Format  requires (lines 92-93).

Line 91 – confusing sentence, perhaps this is an English language issue but need a rewrite for clarity. Suggest dividing to two sentences.

We have divided this sentence in three sentences. We think is much clear now (lines 102-105).

Line 93 – I would recommend removing the specific name of the university. It is not relevant and may create bias. A sentence clarifying that interviews were conducted with the newly graduated nurses and focus groups were conducted with the nursing students would be helpful to the reader.

We have removed the name of the university (line 105). And we have clarified that personal interviews were conducted with NQN and focus group were conducted with 4th-year nursing students (lines 108, 111-113).

Table 1: clear and easy to understand. 

Thank you.

Line 109 – Is this sentence saying that the interview and focus group questions were trialed with 3 nurses and former students? This was unclear to the reader.

We think that we have clarified saying that the interview and focus group were trialed with three nurses and former students of this university (lines 123-124).

Line 141 is redundant. Remove from above or here but does not need to be stated twice.

We have removed a sentence above, under Figure 1 (lines 155-156).

Figure 1 – (line 143) I was confused by the statement that two themes emerged identified as good lecturer and good clinical preceptor. Did these terms come out of the thematic analysis or were they part of the research aim to begin with? The aims should precede the research therefore the statement that these themes emerged from the research makes the aim statement suspect of post hoc manipulation by the research. Perhaps this is just a wording choice issue but needs addressed.  Additionally, it does not make sense that the emerging theme for a good clinical preceptor by the question “What is a good clinical preceptor?” please revise. 

Thank you very much for your insight. As stated in the last paragraph of the Introduction section, these paper is part of a large project about the acquisition of nursing skills and the improvement of the academic and clinical curricula with several specific aims (lines 81-82). The aim presented here was focused on knowing the opinions and perceptions of 4th-year students and NQN about the effectiveness of their academic and clinical trainers. May be we didn´t express ourself correctly.

Our themes came out of the thematic analysis. We have slightly modified the aim of this paper and one of the subthemes (3.4. in line 185) in order to clarify our results. The previous subtheme (What is a good clinical preceptor?) may be confusing in its form, as it was not a question of the interview, it was an emerged subtheme. So, we think this new name is better.

We have never had the intention of manipulate the research.

Line 175 – What is NQN?

NQN is newly-qualified nurses abbreviation. We have clarified for the fist time in line 79.

Line 208 – this is the first mention of the Nursing clinical teacher effectiveness inventory. Did you survey the sample or do interviews/focus groups?  Is this a quantitative study or mixed methods?   The first paragraph of the discussion section does not make sense to the reader.

We have modified the first sentence of the first paragraph. We have clarified that the characteristics and qualities of the "good lecturer” were provided by the NQN and the students of this study. We intend to compare this characteristics provided with the features recorded in the teaching-ability subscale of the Nursing Clinical Teacher Effectiveness Inventory (lines 217-219).

The second paragraph of the discussion section does not seem to go with the study but speaks about the education requirements. If this is significant background information, I suggest moving it to the introduction. It loses the reader focus in the discussion.

Thank you for your recommendation. We have removed this paragraph in order to makes it easier for the reader.

Line 271 – states the generally negative experiences with clinical preceptorships… This is the first time that the idea of negative experiences is introduced. The examples given from the data are not particularly negative and the tone of the manuscript does not identify a particularly negative experience. This needs to be revisited.

Sorry. That has had to be a mistake. We have removed “generally negative” (line 278-279).

The conclusion needs to have a more global significance to the research. As it is written the research is only applicable to the Spanish universities and yet all nursing programs face many of these same challenges seeking qualified educators.

In conclusión section, the last paragraph has been removed to be placed in a “Implication for practice” subheading in Discussion section (lines 266-270). And we have added two last sentences in Conclusion section (lines 278-281).

Reviewer 2 Report

Thank you for the opportunity to review the manuscript “Should we be trained to train? A qualitative approach to good lecturers and good clinical preceptors”. Congratulations to the authors for their work, I found your paper a potentially very valuable resource on Nursing Science and therefore an interesting and relevant contribution to IJERPH.

The manuscript reports on the results the perceptions of newly-qualified nurses and 4th-year Nursing students about what a “good lecturer” and “a good clinical preceptor”.

However, in my opinion there are several aspects should be revised to improve the explanatory power of the manuscript as noted below.

GENERAL COMMENTS:

The authors mentioned that this paper is part of a larger project on the acquisition of nursing skills and the improvement of the academic curricula, and the breadth of the results is such that it cannot be included in a single paper. However, due to the length of the results and the repetition of information in the manuscripts, the authors should better justify this decision.

SPECIFIC COMMENTS:

TITTLE

Correct.

ABSTRACT

In the abstract the results and conclusions seem mixed. It is recommended to structure the information.

INTRODUCTION

In the introduction the authors should better conceptualize the figures of the "good lecturer" and "good clinical preceptor"

METHOD AND RESULTS

Regarding the methodology, the following question arises: Why did the authors not perform participant observation?

Authors should better justify the use of the same data for different publications. This paper repeats a lot of information in the introduction and in the method of another recent publication made by the authors (1).

Much of the material and method section is the same. Tables 1 and 2 are the same in both manuscripts. This aspect concerns me a lot as a reviewer, since the authors must justify very well that they are not performing self-plagiarism.

The author repeat this information in material and method and results section:

“These results are part of a research project on the level of acquisition of skills attained by students of the Degree in Nursing. The breadth and extension of the results obtained could not be covered in one single paper”.

DISCUSSION AND CONCLUSION

The last paragraph of conclusions can´t be deduced from the study:

"Spanish universities and public health systems need preceptorship training programs with assessable objectives for both students of the Nursing degree and clinical preceptors to be able to determine the benefits of such programs in preceptors function throughout their daily activity and in the effectiveness and quality of the students practical training". It is recommended to put the discussion in a section of practical implications of the study.

REFERENCES IN THIS REVISION:

Martínez-Linares, J.M.; Martínez-Yébenes, R.; Andújar-Afán, F.A.; López-Entrambasaguas, O.M. Assessment of nursing care and teaching: A qualitative approach. J. Environ. Res. Public Health. 2019, 16, 2774. doi:10.3390/ijerph16152774.

Author Response

Cover letter – Major revision - Reviewer 2

Thank you very much for the suggested modifications for the paper Should we be trained to train? A qualitative approach to good lecturers and good clinical preceptors. Having them already done, all the authors believe that the article has improved.

Every modification done is described below. In the paper you can find them in green colour:

Thank you for the opportunity to review the manuscript “Should we be trained to train? A qualitative approach to good lecturers and good clinical preceptors”. Congratulations to the authors for their work, I found your paper a potentially very valuable resource on Nursing Science and therefore an interesting and relevant contribution to IJERPH.

The manuscript reports on the results the perceptions of newly-qualified nurses and 4th-year Nursing students about what a “good lecturer” and “a good clinical preceptor”.

However, in my opinion there are several aspects should be revised to improve the explanatory power of the manuscript as noted below.

GENERAL COMMENTS:

The authors mentioned that this paper is part of a larger project on the acquisition of nursing skills and the improvement of the academic curricula, and the breadth of the results is such that it cannot be included in a single paper. However, due to the length of the results and the repetition of information in the manuscripts, the authors should better justify this decision.

Thank you for your suggestions. We have added a sentence in the first paragraph of Materials and Methods section justifying this decisión due to the to the large amount of results and their importance to generate a paper of its own (lines 86-88).

SPECIFIC COMMENTS:

TITTLE

Correct.

Thank you. However the title has been modified as required by reviewer 1 (lines 2-4).

ABSTRACT

In the abstract the results and conclusions seem mixed. It is recommended to structure the information.

We have added a sentence in the Results section of the abstract to distinguish between results and conclusions (lines 22-23). Besides, Conclusions section in Abstract has been modified in order to distinguish both sections (lines 24-26).

INTRODUCTION

In the introduction the authors should better conceptualize the figures of the "good lecturer" and "good clinical preceptor".

We have added a paragraph in Introduction section with definitions of “good lecturer” and “good clinical preceptors” with new references (lines 63-69).

METHOD AND RESULTS

Regarding the methodology, the following question arises: Why did the authors not perform participant observation?

Authors believe that participant observation was not the most appropiate way to collect data to response to the aim of the study. Authors didn´t intend to give their own perceptions on what is a good or bad educators. We only intend to know students and newly-qualified perceptions.

Authors should better justify the use of the same data for different publications. This paper repeats a lot of information in the introduction and in the method of another recent publication made by the authors (1).

In the Introduction section we have included the most important legal framework in which our study is embeded. So, in both article our intention is to frame the context to the reader. Only two referencies are repited in both (references 1 and 2). Reference 2 has been finally removed as a suggestion of changing the Introduction section by reviewer 1.

In the Material and method section we can not changed the content as it was the same. It is indicated in the first sentence on it (line 86-88) and it is justified.  

Much of the material and method section is the same. Tables 1 and 2 are the same in both manuscripts. This aspect concerns me a lot as a reviewer, since the authors must justify very well that they are not performing self-plagiarism.

We can not change Table 1, because participants´ are the same in both paper as they are results of one project. If you prefer, we can remove this table and indicate in the paper that parricipants´ sociodemographic data can be found in the previous publication (Martínez-Linares, J.M.; Martínez-Yébenes, R.; Andújar-Afán, F.A.; López-Entrambasaguas, O.M. Assessment of nursing care and teaching: A qualitative approach. J. Environ. Res. Public Health. 2019, 16, 2774. doi:10.3390/ijerph16152774). 

With regards to Table 2, we have changed the primary questions of the interview guide for those specifically related to the aim of the paper.

We don´t want to perform self-plagiarism. Our intention is to facilitate the reader with the sociodemographic participants´ data.

The author repeat this information in material and method and results section:

“These results are part of a research project on the level of acquisition of skills attained by students of the Degree in Nursing. The breadth and extension of the results obtained could not be covered in one single paper”.

We have removed this information in Results section.

DISCUSSION AND CONCLUSION

The last paragraph of conclusions can´t be deduced from the study:

"Spanish universities and public health systems need preceptorship training programs with assessable objectives for both students of the Nursing degree and clinical preceptors to be able to determine the benefits of such programs in preceptors function throughout their daily activity and in the effectiveness and quality of the students practical training". It is recommended to put the discussion in a section of practical implications of the study.

We have moved this paragraph under an “Implication for practice” subheading at the end of the Discussion section. We think that here fits better (lines 266-270).

REFERENCES IN THIS REVISION:

Martínez-Linares, J.M.; Martínez-Yébenes, R.; Andújar-Afán, F.A.; López-Entrambasaguas, O.M. Assessment of nursing care and teaching: A qualitative approach. J. Environ. Res. Public Health. 2019, 16, 2774. doi:10.3390/ijerph16152774. 

Round 2

Reviewer 1 Report

I recommend accepting the manuscript as revised.

Reviewer 2 Report

Congratulations to the authors for the changes introduced in the manuscript.

I give my approval for publication.

Best regards.